# In Vitro Study of Shear Bond Strength in Direct and Indirect Bonding with Three Types of Adhesive Systems

**DOI:** 10.3390/ma13112644

**Published:** 2020-06-10

**Authors:** Angelica Iglesias, Teresa Flores, Javier Moyano, Montserrat Artés, Francisco Javier Gil, Andreu Puigdollers

**Affiliations:** 1Department of Orthodontics, Universitat Internacional de Catalunya, 08195 Barcelona, Spain; angelica@uic.es (A.I.); mflorfra@uic.es (T.F.); jmoyano@uic.es (J.M.); martes@uic.es (M.A.); 2Bioengineering Institute of Technology, Universitat Internacional de Catalunya, 08195 Barcelona, Spain; xavier.gil@uic.cat

**Keywords:** self-etch, self-adhesive, indirect bonding, shear bond strength

## Abstract

This study aimed to compare the shear bond strength (SBS) and adhesive remaining index (ARI) using one conventional and two novel adhesive systems with clinical step reduction and direct and indirect bonding. A sample of 72 human premolars were divided into six groups of 12 samples. The first three groups (G1, G2, G3) were bonded with a direct technique, while the remaining groups (G4, G5, G6) were bonded by the indirect technique. Groups G1 and G4 used conventional acid-etching primer composite (XT); groups G2 and G5 used self-etching bonding (BO), and groups G3 and G6 had an acid-etching treatment followed by a self-adhesive composite (OC). All groups were exposed to thermocycling. Shear bond strength was analyzed with a universal test machine, and the ARI was examined with 4× magnification. The results showed statistically significant differences between the three adhesive systems. The highest strength values were observed in the XT group G1 (13.54 ± 4 MPa), while the lowest were shown in the BO G2 samples (5.05 ± 2 MPa). There was no significant difference between the direct or indirect bonding techniques on the three compared groups. The type of primer and bonding material significantly influenced the SBS. Values with self-etching bonding were below the minimum recommended for clinical use (5.9–7.8 MPa). There was no difference between indirect and direct bonding techniques. The lowest ARI scores (0–1) were observed in both self-etching and BO groups. Further clinical studies are needed to compare in vivo results.

## 1. Introduction

Since the introduction of the first bonding systems, there has been a constant effort to improve the quality of materials [1]. Researchers [2,3] have developed new adhesives based on the need to increase shear bond strength (SBS), decrease bonding time, achieve an efficient reduction of the clinical bonding steps, and preserve the enamel. Bond strength should be of an optimum force rather than too much or too little [2]. According to Reynolds [4], the minimum bond strength should be in the range of 5.9–7.8 MPa to withstand masticatory forces. Excessive bond strength forces (greater than 40–50 MPa), increase the risk of enamel damage during debonding and should be avoided; while bond failures during treatment are a consequence of insufficient bond strength values and are also not desirable [2,5]. Therefore, the nature of the adhesive is of great importance in regard to the bond strength, composite left on teeth, and enamel injury [6]. Enamel loss at debonding reaches up to 120.6–189.98 µm. An additional 10–30 µm is affected by the previously performed acid-etching process, and 55.6 µm [7,8] is lost in the removal of the remnant adhesive after debonding. It could be assumed that these values multiply every time a bracket is re-bonded [2].

Conventional adhesive systems use three different agents during the bonding process: an enamel conditioner, a primer solution, and an adhesive resin. Phosphoric acid has remained the primary enamel etchant since its introduction by Buonocore [9], despite the enamel loss previously described. New etching systems have tried to solve this problem, adding fluoride releasable components to protect the enamel, and using adhesive cements [10] or developing less detrimental components for the etching of the enamel by a combination of conditioning (Phenyl-P) and priming (HEMA and dimethacrylate) agents into a single acidic primer solution. These so-called self-etching primers can be used on enamel and dentin in a single step [11]. Numerous comparative studies between conventional and self-etching adhesives have been performed to determine the effectiveness of adherence in both techniques [12,13]. In addition, some of the studies claim that the gentler components of self-etching primers produce less enamel loss than phosphoric acid [14,15].

Indirect bonding (IB) has emerged in recent years as the best option to achieve a precise bracket placement. The technique introduced was by Silverman [16], and modified by Thomas [17], and has become the basis of current indirect bracket bonding methods. It consists of positioning the bracket in a laboratory working cast, followed by the fabrication of a transference tray to assure correct bonding in the patient. With a direct vision of the cast model, accurate placement of the bracket, less chair time [18], less patient discomfort, and improved ability to bond posterior teeth are some of the advantages that have been described [19,20,21]. Additionally, due to prior bracket placement in the laboratory, it has been suggested that IB allows more accurate bracket positioning [17,21,22].

Classification of samples according to the adhesive remnant index (ARI) reveals information about the bracket bonding enamel interface and has great value concerning the enamel condition after debonding. Lower values tend to show less damage on the surface of the enamel and are rarely related to enamel fractures, but can be related to insufficient bond strength. On the contrary, higher ARI values are shown in efficient unions between adhesive and enamel, but are often associated with greater surface damage [23].

A novel bonding protocol involving new adhesive systems that reduce steps during bonding, such as self-etching or self-adhesive methods, is of great clinical interest among orthodontists. It would also be of interest to incorporate the indirect bonding technique, given its previously mentioned advantages. Therefore, the aim of this in vitro study was to evaluate the efficiency of three types of bonding systems: a conventional acid-etching composite, a self-etching adhesive method, and a new self-adhesive composite, in combination with two bonding techniques (direct and indirect bonding). The efficiency of the systems was evaluated by computation of the shear bond strength (SBS) and the adhesive remnant index (ARI), aiming to obtain information that helps to develop the best adhesion protocol for further clinical studies.

## 2. Materials and Methods

### 2.1. Sample Size

A preliminary pilot study was conducted to test the methodology. After pilot study analysis, sample size calculation was set to achieve a statistical power of 80% with a significance level of 5%, taking into account a 2.5 standard deviation. A sample of 12 premolars per group was estimated. With this data, a study protocol was developed and approved by the ethics committee at the Universitat Internacional de Catalunya (Sant Cugat, Barcelona, Spain).

### 2.2. Sample Preparation

Seventy-two freshly extracted premolars showing no decay or fractures were introduced into 0.9% physiological saline and stored at 37 °C. The three tested bonding materials are shown in Table 1. All samples were cleaned, and organic residues were removed with gauze. The bonding surface was abraded with a cup and fluoride-free prophylaxis paste (Detartrine, Septodont, Saint-Maur-des-Fossés, France) at low speed, applied by a handpiece (10 s), and posteriorly washed with water and dried with oil-free compressed air. Afterward, the sample was randomly divided into 6 groups according to the type of adhesive and bonding technique (Table 2), with 12 premolars per group.

All samples were mounted with a custom-made jig, as described by Flores et al. [12], to standardize the position of the teeth. The buccal surface of the teeth was set parallel to the direction of the shear during mechanical debonding tests. Samples of the direct bonding groups were mounted individually, and the indirect bonding samples were mounted in groups of 4 teeth per block (Figure 1). For the indirect bonding groups, a cast of the block-mounted units was made with a silicone impression (Hydrorise Putty Fast, Zhermack, Marl, Germany), and hard stone (Elite Model, Zhermack, Marl, Germany) was poured into the impression. The cast was burnished with a separator (Prothyl Isolator, Zhermack, Marl, Germany) and left to dry at ambient temperature for 1 min. The next step was the positioning of the premolar brackets (Victory Series, 3M Unitek, Monrovia, CA, USA) [12].

Group 1: TX/Direct Bonding

The teeth were pre-treated with 37% orthophosphoric acid for 30 s and irrigated with water for 10 s before the adhesion procedure of the brackets. The bonding protocol was followed according to the manufacturer’s instructions. Proper drying of the enamel after etching was performed, followed by the application of a thin layer of Transbond XT primer (3M Unitek, Monrovia, CA, USA) on the tooth surface. Then, a small amount of composite was dispensed in the mesh, and bracket positioning was undertaken with a constant low pressure and removal of excess material. Finally, it was polymerized for 30 s (10 mesial, 10 distal, and 10 occlusal) with a LED lamp (Bluephase, Ivoclar Vivadent AG, Schaan, Liechtenstein).

Group 2: BO/Direct bonding

In this group, we used a self-etching composite based on a single bottle primer presentation and the composite Beauty Ortho Bond II (Shofu, Kyoto, Japan). It enabled us to do the priming and etching in a single step, following the manufacturer’s instructions. The self-etching primer was applied on the enamel surface by rubbing for 3 s without previous acid-etching. Then, the surface was dried with low pressure, and the bracket was placed using the Beauty Ortho Bond II light-cured composite by direct bonding. The photopolymerization was done in the same way as that in group 1.

Group 3: OC/Direct Bonding

The teeth were pre-treated with 37% orthophosphoric acid for 30 s and irrigated with water for 10 s before the adhesion procedure of the brackets. Proper drying at a low speed was carried out, and we proceeded to spread the self-adhering composite GC Ortho Connect onto the bracket mesh. The positioning of the bracket was carried out with a low and constant pressure and posterior removal of excess material. Finally, we proceeded with photopolymerization, using an LED light (Bluephase, Ivoclar Vivadent AG, Schaan, Liechtenstein) for 30 s (10 mesial, 10 distal, and 10 occlusal).

Group 4: TX/Indirect bonding

For the indirect bonding, we obtained cast models as previously described, in which marks were made for the positioning of the bracket. The bracket was then positioned with Transbond XT and LED light-curing of 20 s per bracket. Once the brackets were placed in the cast, we continued the fabrication of the transference tray with Elite Glass (Zhermack, Marl, Germany) that was applied to the cast by a mixing pistol in a uniform way over the teeth with a work time of 40 s and 2 min 15 s of setting time in a 23 °C temperature. Once we had obtained the transference tray, we removed it from the cast with the bracket and proceeded with the sandblasting.

For the treatment of the natural teeth, 37% orthophosphoric acid was applied for 30 s, posterior irrigation with water occurred for 10 s, the drying of the surface was carried out as in previous groups, and the application of the primer followed the manufacturing instructions. We then placed a fine layer of composite in the bracket mesh and made the transference tray with light-curing for 30 s (Figure 2).

Group 5: BO/Indirect Bonding

In this group, we applied the same protocol to natural teeth as in group 2, and the transference tray was made as described in group 4.

Group 6: OC/indirect group

In this group, we applied the same protocol to natural teeth as in group 3, and the transference procedure was the same as described in group 4.

### 2.3. Thermocycling

The tooth samples were stored with hydration in a physiological saline solution of 0.9% at 37 °C. All groups were exposed to thermocycling before debonding 24 h after bracket placement. The thermocycling consisted of 1500 cycles in a water bath at 5 and 55 °C, with bath times at 1 min intervals.

### 2.4. Debonding Resistance Test

We tested the samples of all the different groups with a universal testing machine (Quasar 5, Galdabini, Cardano al Campo, Italy) at a maximum speed of 0.1 mm/min [12]. The result of the force was obtained in newtons and then converted to MPa, dividing the force between the bracket area (9.75 mm^2^).

### 2.5. Adhesive Remnant Index

Examples of adhesive remnant index (ARI) classification and values are shown in Figure 3. Each sample was evaluated and photographed by one operator (A.I.) with a microscope at 4× magnification, and was classified based on a visual scale in which scale 0 corresponded to no adhesive remaining; 1 corresponded to less than half of the adhesive remaining; 2 corresponded to more than half of the adhesive remaining; 3 corresponded to all adhesive remaining [24].

### 2.6. Statistical Analysis

Numerical variables were described with mean and standard deviation, while the median was described with minimum and maximum values. Categorical variables were described with frequencies and percentages.

Differences of SBS between the groups of the study were analyzed with one-way ANOVA, and the Shapiro–Wilk test was used to check the normality of residuals. In addition, a comparative post hoc test was used with Bonferroni correction.

The Kruskal–Wallis and Mann–Whitney tests were used to analyze differences in the distributions of the ARI scores.

The samples were classified based on a visual scale by one operator (A.I.), and intra-operator analysis was performed for the ARI values with a regression line of all the samples. The validity of the classification was confirmed with an 0.85 correlation coefficient.

Results were analyzed with a statistical software program (Statgraphics, Warrenton, VA, USA). The level of statistical significance was set at *p* < 0.05.

## 3. Results

### 3.1. Shear Bond Strength

A description of the results of the shear bond strength is shown in Table 3 The highest strength values were observed in the acid-etching groups (OC and XT materials), with the XT direct group (13.5 ± 4 MPa) having the highest values. The lowest values were shown in the BO samples, with the indirect group (5.1 ± 2 MPa) representing the lowest scores (Figure 4).

One-way ANOVA showed significant effects of groups (*p* < 0.001), and we checked the normality residuals with the Shapiro–Wilk test (*p* = 0.103).

The difference between groups and multiple comparisons by t-test with Bonferroni correction are shown in Table 4, in which all acid-etching groups (XT, OC) demonstrate significant difference when compared with the self-etching groups (BO).

### 3.2. Adhesive Remnant Index

The ARI analysis by materials was performed by a Kruskal–Wallis test that showed statistically significant differences between groups (*p* < 0.001). Thus, a subsequent Mann–Whitney test with Bonferroni correction was performed to analyze the differences between them (Table 5). The results showed a significant difference between the acid-etching groups (XT, OC) and the self-etching group (BO). The XT and CT groups showed a majority of scores between 2 and 3, while BO groups showed predominant values of 0–1 (Figure 5). There was no difference between the bonding techniques (direct and indirect bonding).

Group 1 exhibited the highest ARI scores, which represented the most composite remaining on the enamel surface, while the lowest ARI score (least composite remaining) was observed in Group 5.

## 4. Discussion

This investigation intended to analyze whether the reduction of clinical steps in the two bonding materials had an influence on SBS results when compared to the conventional etch bonding resin procedure in both direct and indirect bonding procedures. Moreover, we also compared how ARI can vary with the use of different bonding systems and bonding techniques to evaluate enamel conditions and make assumptions about which step of the bonding procedure may cause failures in adhesion.

The limitations of any in vitro study includes variables of the intra-oral environment such as saliva contamination, enamel composition, or occlusal forces, that greatly influence adhesion and can alter the results of similar in vivo studies. We tried to mitigate these differences by thermocycling all samples up to 1500 cycles [12]. Even so, the conclusions of an in vitro study cannot be directly extrapolated to the clinical environment. As such, it is important to reproduce these parameters in further randomized clinical trials and then compare the results with the present study. Moreover, other very common intra-oral situations, such as the presence of zirconia prosthetics, the demand for aesthetic ceramic brackets, or the influence of external conditions like magnetic resonance exposition, that have been reported to influence SBS could not be evaluated in our in vitro study [25,26,27,28]. However, further studies can be designed to check the behavior of the materials in these common clinical situations.

All 72 premolars were randomly allocated to the six groups of study. Even though we were not able to perform intra-operator tests for the SBS trials, all debonding mechanical tests were performed by an experienced orthodontist in this field (A.I.) and supervised by a senior researcher in bonding materials (T.F.). Premolars are the most commonly used teeth for this kind of study as their availability is not in conflict with the ethics committee due to frequent extraction for therapeutic orthodontic reasons. In addition, it is advisable to use the same teeth, since the area of the bracket mesh is necessary to obtain the MPa conversion, and having several types of bracket widths could be a variable to take into account.

We compared three different bonding materials in which the conventional acid-etching primer composite (XT) served as a control group to compare the two other novel bonding types (BO and OC). As far as we know, the self-etching system used in our study has not been commercialized and was loaned for testing in the university without any commercial interest. At the time of this study, there was no literature concerning the self-adhering system we used either. From the clinical point of view, any reduction of steps represents an improvement of the technique, reducing potential mistakes by the operator between steps, and reducing chair time. For this reason, we found the new bonding materials to be of interest for this research.

The indirect bonding technique emerged as an alternative to reduce operator stress during chair time and achieve a more accurate placement of the bracket from the beginning of the treatment [29,30,31] Re-bonding brackets is an enamel endangering procedure that is commonly practiced in orthodontics. The process of bracket replacement increases the enamel loss previously mentioned in initial bonding [7,32] as well as in the new pre-treatment of the enamel and the posterior final debonding. On this matter, the indirect bonding technique showed no statistically significant difference in any of the groups when compared with its direct bonding analogs, indicating it is an optimal bonding technique [13,33,34,35,36]. Although other authors reported significantly lower values of SBS in indirect bonding samples when compared with direct bonding [12,37], it must be pointed out that many studies are difficult to compare because of the numerous steps. The lack of standardization in indirect techniques adds variables concerning the material of the trays [38,39,40], the adhesion system (light or chemical cured), and even the type of brackets and positioning guides (slot heights, center of crown, parallelism guide) [41].

The bond strength of orthodontic appliances must be able to withstand intra-oral forces during treatment. An optimal adhesion with SBS minimum values of 5.9–7.8 MPa [4] must be reached independently of the type of material or the bonding technique. The Beauty Ortho Bond II (BO), eliminated the two-step presentation of primers A and B (conditioning and priming) of its predecessor into a single bottle mixture of the monomer primer. However, this material showed values of SBS that were below the minimum necessary forces previously described [4], and significantly lower than the acid-etching groups. No other literature reference was found to analyze a correlation or establish where the failure might have occurred. Its predecessor, the Beauty Ortho Bond, has been widely studied as one of the standards for self-etching orthodontic materials in comparison with conventional bonding systems [3,8,12,42,43,44,45]. Therefore, it could be speculated that the bonding failure occurs at some point in the chemical reaction of the self-etching new “all in one” bottle presentation, and should compel the commercial brand to conduct more trials before its public sale.

The self-adhering resin simplifies the bonding process by reducing it to a one-step procedure after enamel etching. Previous studies with self-adhering flowable resins stated significantly lower SBS values when compared with conventional systems [2,46,47]. These initial studies tried out composites originally intended for restorative procedures instead of orthodontic purposes, which had a more liquid composition. Contrarily, the self-adhesive material used in this study has been manufactured for orthodontic use and has demonstrated effective bond strength at debonding, with similar values to the conventional acid-etching primer composite (XT) samples in agreement with previous studies [48].

Data about failure at the bracket-adhesive or the adhesive-enamel interface has both advantages and disadvantages. The tested samples revealed significantly lower values of ARI in both BO groups, in agreement with previous studies [8,12,32]. The majority of the acid-etching samples (G1, G3, G4, G6) exhibited ARI scores between 2 and 3. When enamel surface remains relatively intact after debonding, considerable chair time is needed to remove the residual adhesive, with the risk of damaging the enamel substance also present in the cleaning procedure [32,49]. This issue can be minimized by alternative removal procedures that are less detrimental to enamel, such as sandblasting [50,51]. In the case of a failure in the enamel adhesive interface, less adhesive will remain, but the enamel surface can also be damaged when a failure occurs in this way, even producing enamel fractures [27,52]. The acid-etching group reflected an optimal union between enamel and adhesive, with high ARI values, which could also justify the higher SBS results. Meanwhile, the lower ARI values, which also corresponded to the lower SBS results, reflected a bond failure or insufficient union between the adhesive and enamel.

## 5. Conclusions


All self-etching samples showed significantly lower shear bond strength values when compared with the acid-etching groups. BO samples showed values below what is recommended for clinical use (5.9–7.8 MPa).The self-adhesive composite groups demonstrated similar values to those of the XT that acted as a control group, illustrating that this material had efficient forces, regardless of the reduction of one clinical step.There were no statistical differences between the indirect bonding technique groups when compared with their direct bonding analogs. Therefore, indirect bonding technique is a valid alternative bonding procedure.ARI scores suggested that the acid-etching samples (with the majority of scores between 2 and 3) created an effective bonding-enamel attachment but could lead to further enamel damage during the polishing procedure.Further clinical studies with the same materials are necessary.


## Figures and Tables

**Figure 1 materials-13-02644-f001:**
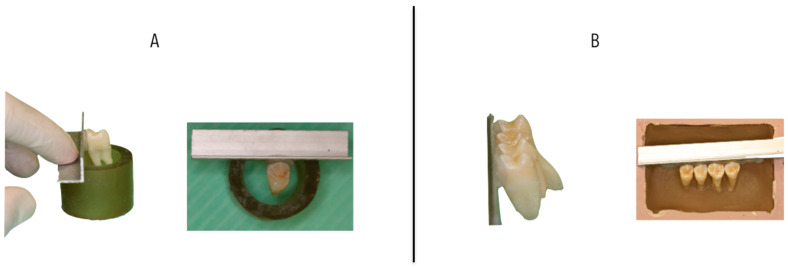
Representative images of mounting jigs. (**A**) Direct bonding samples, (**B**) Indirect bonding samples.

**Figure 2 materials-13-02644-f002:**
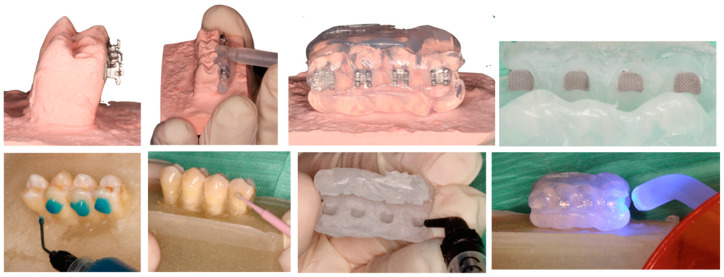
Representative images of indirect bonding samples steps (group 4).

**Figure 3 materials-13-02644-f003:**
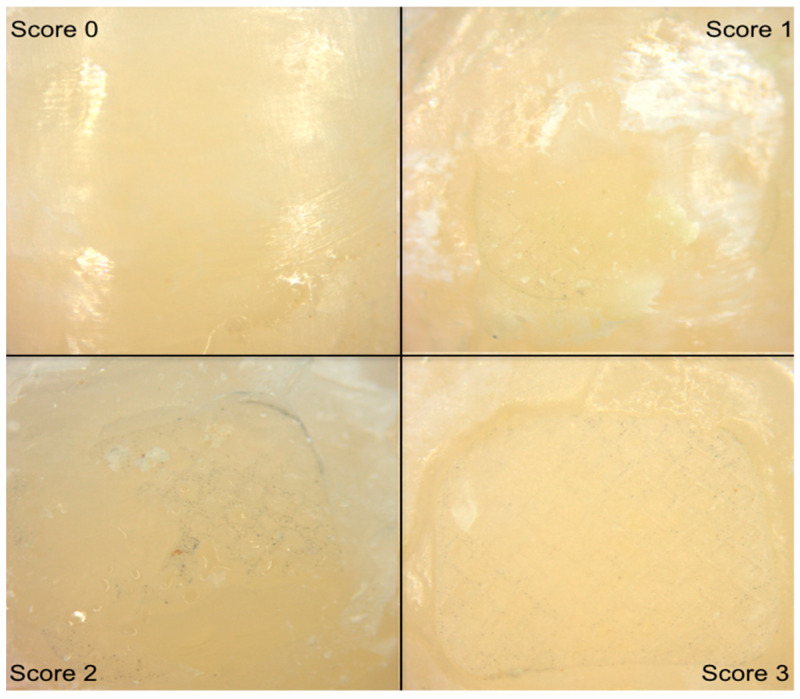
Representative images for adhesive remnant index (ARI) under microscopic examination. ARI 0 (no adhesive remaining); ARI 1 (less than half of the adhesive remaining); ARI 2 (more than half of the adhesive remaining); ARI 3 (all adhesive remaining).

**Figure 4 materials-13-02644-f004:**
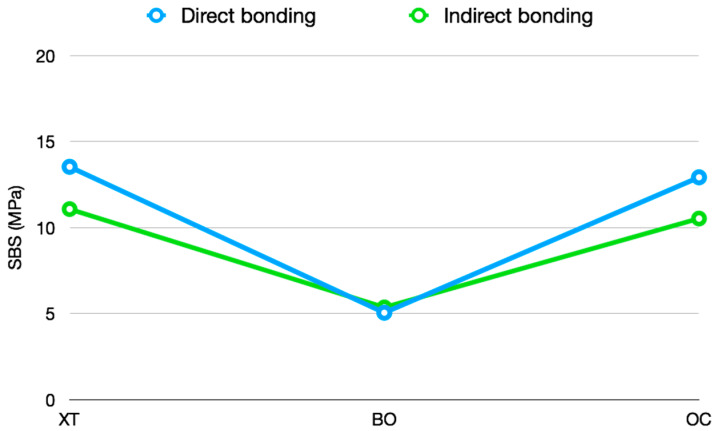
Mean of bonding technique interactions according to the levels of the factors material and bonding technique.

**Figure 5 materials-13-02644-f005:**
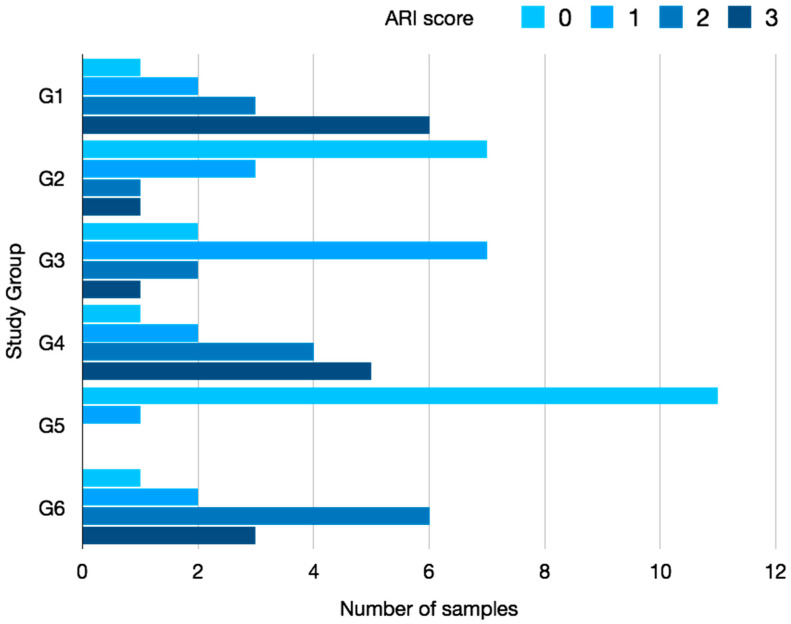
Bar diagram of ARI classification according to groups.

**Table 1 materials-13-02644-t001:** Composition of tested adhesive systems.

Bonding Material	Manufacturer	Components	Composition
**XT** Transbond XT	3M Unitek, Monrovia, California, USA	Etching Gel Primer Paste	37% phosphoric acid, tetraethyleneglycol 1.39 dimethacrylate (TEGDMA), bisphenol A diglycidyl methacrylate (BIS-GMA); Bis-GMA, TEGDMA, silane-treated quartz, amorphous silica, camphorquinone
**BO** Beauty Orthobond II	Shofu, Kyoto, Japan	Primer A (self-etching) Paste	Water, solvent, phosphoric acid monomer, ethanol, TEDGMA, surface pre-reacted glass-ionomer, filler, Bis-GMA, camphorquinone
**OC** GC Ortho Connect	GC America, Alsip, Illinois, USA	Etching Gel Paste (self-adhesive)	37% phosphoric acid (7.7.9-trimethyl-4.13-dioxo-3.14-dioxa-5.12-diazahexadecane-1.16-diyl bismethacrylate)

**Table 2 materials-13-02644-t002:** Study design of the groups to compare the shear bond strength (SBS) and the adhesive remaining index (ARI) using three adhesive systems and two bonding techniques (direct and indirect bonding).

Bonding Protocol	G1	G2	G3	G4	G5	G6
Adhesive	XT	BO	OC	XT	BO	OC
Bonding Technique	Direct	Direct	Direct	Indirect	Indirect	Indirect

**Table 3 materials-13-02644-t003:** Descriptive values of shear bond strength for the overall sample, levels of two factors, and the six study groups.

Comparison	Mean (sd)	Median [Min–Max]
Overall	9.7 (4.5)	9.4 [2.7–20.2]
Material	–	–
XT ^a^	12.3 (4.1)	12.5 [4.8–20.2]
BO ^b^	5.2 (1.6)	5.1 [2.7–8.7]
OC ^a^	11.7 (3.4)	11.7 [5.1–17.8]
BT	–	–
Direct	10.5 (4.9)	11.7 [2.7–20.2]
Indirect	8.9 (3.9)	8.2 [3.3–18.6]
Material:BT	–	–
(G1) XT—Direct	13.5 (4.0)	13.7 [4.8–20.2]
(G2) BO—Direct	5.1 (2.0)	4.5 [2.7–8.7]
(G3) OC—Direct	12.9 (3.0)	13.3 [5.6–17.8]
(G4) XT—Indirect	11.1 (3.9)	9.6 [6.9–18.6]
(G5) BO—Indirect	5.4 (1.2)	5.4 [3.3–7.4]
(G6) OC—Indirect	10.5 (3.4)	9.9 [5.1–17.3]

Different letters (^a, b^) show significant statistical difference.

**Table 4 materials-13-02644-t004:** Comparisons of SBS variable for levels of two factors and six study groups.

Comparison	Difference	CI 95%	*p*-Value
Material	–	–	–
XT–BO	7.10	5.26; 8.94	<0.001
OC–BO	6.52	11.73; 8.10	<0.001
XT–OC	0.58	−1.61; 2.77	1
BT	–	–	–
Direct–indirect	1.51	−0.60; 3.63	0.160
Material: BT	–	–	–
G1–G5	8.17	5.56; 10.79	<0.001
G1–G6	3.01	−0.16; 6.18	0.316
G1–G4	2.46	−0.91; 5.83	0.871
G2–G3	−7.88	−10.08; −5.67	<0.001
G2–G1	−5.49	−11.23; −5.74	<0.001
G2–G5	−0.31	−1.75; 1.12	1
G2–G6	−5.48	5.05; 10.53	0.001
G2–G4	−6.03	−8.74; −3.32	0.002
G3–G1	−0.61	−3.63; 2.42	1
G3–G5	7.57	5.54; 9.59	<0.001
G3–G6	2.40	−0.35; 5.15	0.958
G3–G4	1.85	−1.15; 4.84	1
G5–G6	−5.16	−7.43; −2.89	0.002
G5–G4	−5.72	−8.30; −3.14	<0.001
G6–G4	−0.55	−3.69; 2.59	1

**Table 5 materials-13-02644-t005:** Frequency and percentage for overall sample, two factors, and six groups.

Comparison	0	1	2	3	*p*-Value
Overall	24 (32%)	18 (24%)	16 (21.3%)	17 (22.6%)	
Material	–	–	–	–	<0.001
XT ^a^	2 (8%)	4 (16%)	7 (28%)	12 (48%)	–
BO ^b^	19 (76%)	4 (16%)	1 (4%)	1 (4%)	–
OC ^a^	3 (12%)	10 (40%)	8 (32%)	4 (16%)	–
BT	–	–	–	–	0.978
Direct	11 (28.2%)	13 (33.3%)	6 (15.4%)	9 (23.1%)	–
Indirect	13 (36.1%)	5 (13.9%)	10 (27.8%)	8 (22.2%)	–
Material:BT	–	–	–	–	–
(G1) OC—Direct	2 (15.4%)	8 (61.5%)	2 (15.4%)	1 (7.7%)	–
(G2) BO—Direct	8 (61.5%)	3 (23.1%)	1 (7.7%)	1 (7.7%)	–
(G3) OC—Direct	2 (15.4%)	8 (61.5%)	2 (15.4%)	1 (7.7%)	–
(G4) XT—Indirect	1 (8.3%)	2 (16.7%)	4 (33.3%)	5 (41.7%)	–
(G5) BO—Indirect	11 (91.7%)	1 (8.3%)	0 (0%)	0 (0%)	–
(G6) OC—Indirect	1 (8.3%)	2 (16.7%)	6 (50%)	3 (25%)	–

Different letters (^a, b^) show significant statistical difference.

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
