# Peer review of "In Vitro Study of Shear Bond Strength in Direct and Indirect Bonding with Three Types of Adhesive Systems"

_materials, 2020, doi:10.3390/ma13112644_

Round 1
Reviewer 1 Report
General comments:
I read this article with interest and, in my opinion, it does not provide any interesting information. The style is that of a student's essay, at most.
The overall impression is that this text was assembled in a great hurry, with no attention to details or language. Such a low quality presentation is not suitable for a journal with an IF of almost 3, as Materials.
I truly can't imagine if anyone would be interested in reading about constant pressure while cementing, removing excess material before light-curing and many other very well known issues.
The text must be thoroughly checked for proper english.
No images are presented (of the samples, cast, tray, etc., during testing etc). Only 2 graphs in the results section, not enough to document the work.
The conclusion only enumerates some observations. No pertinent conclusion has been drawn.
Punctual comments:
Give full for SBS in the title.
Remove dot after title.
Rector is not an affiliation.
The corresponding author has no affiliation.
Remove chapters from abstract.
TX or XT group? Please use one.
Please arrange text according to indications for authors and template.
line 41. No need for Mega Pascal (Mpa). MPa is the correct form.
line 67. Please give manufacturer and location for the products.
Please modify tables according to template.
Please explain table 2. There is no reference in the text. You can not just display a table following a subsection. Same for table 3 and 4. Also Figure 2.
Please number subsections and use the proper font style.
line 112. "one primer (Primer A and Primer B in one bottle presentation)". This is not an admissible expression in a "scientific" paper.
line 114. "This product contains one bottle of adhesive". Same as above.
line 120. "The teeth were prepared with a previous prophylaxis like the first group". This is only of example of faulty language. There are too many to enumerate.
line 156, 161. The dot belongs in the end of a sentence, not before one.
In my opinion this is not suitable to be published in Materials, even if a revised version is provided. My advice would be to make the necessary corrections and try to submit it to another journal.
Author Response
Reviewer one
Dear reviewer, we are very grateful for all the comments and the thorough examination. We have make major improvements in the original manuscript thanks to your observations and we kindly ask you to reconsider our paper for this special issue on the journal. As you have mentioned, there are studies that have been conducted on this topic. But, as far as we know, there is no a single study that compares resistant to debonding performance among the standard bonding material (XT) with this two materials that present interesting benefits from the clinical point of view with the reduction of steps of the bonding process. Therefore, it is of great importance to prove these new adhesion systems to be a reliable material before its clinical use. Thus, we think that for every novel adhesion system, it is of most importance that both in vitro and in vivo studies take place and validate its liability.
General comments:
C: I read this article with interest and, in my opinion, it does not provide any interesting information. The style is that of a student's essay, at most.
A: Major improvements of the text have been made to highlight what we find to be interesting information.
PD: Please find attached the revised version of the manuscript
C: The overall impression is that this text was assembled in a great hurry, with no attention to details or language. Such a low quality presentation is not suitable for a journal with an IF of almost 3, as Materials.
A: We apologize for the faulty language and details missteps. Major changes have been made in the manuscript specially regarding this points to achieve journal´s standard
C: I truly can't imagine if anyone would be interested in reading about constant pressure while cementing, removing excess material before light-curing and many other very well known issues.
A: We honestly believe that, as we explained at the introduction of this answer, that the two novel bonding materials tested are of interest from the clinical point of view and needed a first in vitro study that will be followed by a clinical study.
C: The text must be thoroughly checked for proper English. No images are presented (of the samples, cast, tray, etc., during testing etc.). Only 2 graphs in the results section, not enough to document the work.
A: Figures1 and 2 have been added to portray laboratory steps and sample manipulation
C: The conclusion only enumerates some observations. No pertinent conclusion has been drawn.
A: The conclusions have been re-stated to present a clearer message of the results
C: Give full for SBS in the title.
A: This point have been corrected
C: Remove dot after title.
A: Done
C: Rector is not an affiliation.
A: This point have been corrected
C: The corresponding author has no affiliation.
A: All affiliations have been corrected
C: Remove chapters from abstract.
A: All subheadings have been removed from the abstract
C: TX or XT group? Please use one.
A: XT is the correct abbreviation. This error has been corrected
C: Please arrange text according to indications for authors and template.
A: Template has been followed at all times using mpdi word template
C: Line 41. No need for Mega Pascal (Mpa). MPa is the correct form.
A: We have change to MPa in all cases (Line 47)
C: Line 67. Please give manufacturer and location for the products.
A: This information has been added to table 1 and in further product mentions in the material and methods section
C: Please modify tables according to template.
A: Tables have been modified with the word table tool as indicated in the journals´ instructions for authors
C: Please explain table 2. There is no reference in the text. You can not just display a table following a subsection. Same for table 3 and 4. Also Figure 2.
A: All tables and figures have been accordingly mentioned in the text.
C: Please number subsections and use the proper font style.
A: Subsections have been properly numbered
C: Line 112. "one primer (Primer A and Primer B in one bottle presentation)". This is not an admissible expression in a "scientific" paper.
A: This expression has been substituted for proper description. (Line 326)
C: Line 114. "This product contains one bottle of adhesive". Same as above.
A: This point has also been corrected.
C: Line 120. "The teeth were prepared with a previous prophylaxis like the first group". This is only of example of faulty language. There are too many to enumerate.
A: This and other corrections have been made in regards of language
C: Line 156, 161. The dot belongs in the end of a sentence, not before one.
A: This point has been corrected
C: In my opinion this is not suitable to be published in Materials, even if a revised version is provided. My advice would be to make the necessary corrections and try to submit it to another journal.
A: Major correction work has been made to the manuscript to achieve the Materials Journal´s standards. We kindly ask for your consideration to be published in the special issue in which we originally submitted the manuscript.

Reviewer 2 Report
Dear authors,
I would like to thank you for your work on this research project. The project is interesting and has potential merit for publication. However, I have several methodological concerns that need to be addressed. Furthermore, attention is needed for grammatical corrections/English editing throughout the text.
Please consider the following points:
Title: Please avoid abbreviations in the title; and change “3” to “three”
Abstract
Line 21: Please note: sentences should not start with a number; 72: please change to Seventy-two
Line 21: Please clarify number of teeth in each group (were the premolars equally divided?)
Lines 21-23: Sentence needs grammatical correction
Lines 23-24: I think parentheses should be removed
Line 26: Abbreviations should not be used in the start of a sentence
Line 28: Please keep abbreviations consistent: TX versus XT group
Introduction
Line 38: needs grammatical correction
Line 46: Please change assume to past tense (assumed)
Lines 55-56: Note: periods missing at end of sentence
Line 57-61: sentences need grammatical correction
Line 66: please correct “these” to “this” alternative bracket placement procedure. Similarly, line 67
Materials and Methods
Lines 81-82 need grammatical correction
Line 82: Please note typo: “Fort he”
Please define abbreviations in the Tables
Line 147: please correct grammatically
Furthermore, please note the following major methodological concerns:
There is no mention of how allocation of teeth to the study groups were performed (for instance ideally there should have been randomization).
Ideally, assessment of the outcome should have been blinded to minimize biases. The lack of randomization and blinding should be addressed; and limitation should be presented in the Discussion section.
Statistical analysis
Since multiple group comparisons were performed; post hoc testing (such as Bonferroni test) is needed after ANOVA to assess differences between each group pair and to control errors.
Also, what is the reliability in the measurements? Were the intra- and inter-observer reliabilities examined?
Furthermore, how was the sample size estimated? Ideally, a priori power analysis is needed to calculate the required sample size. Lack of a power analysis is a major limitation.
Results
Pairwise comparisons need to be performed and p-values need to be adjusted when ANOVA is significant. For instance, it is mentioned that: “The results showed statistically significant differences between the three types of adhesive systems.” Further testing is needed for group-pair comparisons; and adjusted p-values need to be provided for each group-pair comparison.
Similarly, detailed p-values need to be reported for all outcomes including the adhesive remnant index.
Discussion
Line 220: Sentence needs grammatical correction.
Lines 243-4: Please note that grammatical correction is needed.
Line 283: needs grammatical correction
Need to further address potential limitations; such as lack of power, potential biases and error in measurements. Also, since only premolars were examined this could affect generalizability of the results. The generalizability and clinical implications of this study need to be addressed.
Conclusions:
Needs grammatical correction, also please use abbreviation as used previously in the text.
Author Response
Dear reviewer
We want to thank you for your thorough revision of our manuscript and much appreciated comments. We have made a major effort in answering all methodological issues to achieve the journals excellence standards. We hope we addressed all your concerns in this new carefully revised version of the manuscript. Note also that some major modifications in regards of grammar have been made thanks to your observations. .
Title
C: Please avoid abbreviations in the title; and change “3” to “three”
A: The title has been corrected
Abstract
C: Line 21: Please note: sentences should not start with a number; 72: please change to Seventy- two
A: This point has been corrected
C: Line 21: Please clarify number of teeth in each group (were the premolars equally divided?)
A: Premolars were randomly divided into 6 groups of 12 teeth per group. This has been clarified in the text.
C: Lines 21-23: Sentence needs grammatical correction
A: We made all grammatical corrections requested
C: Lines 23-24: I think parentheses should be removed
A: We removed some of the parentheses in the abstract section
C: Line 26: Abbreviations should not be used in the start of a sentence
A: We have corrected abbreviation.
C: Line 28: Please keep abbreviations consistent: TX versus XT group
A: XT is the correct abbreviation. This point has been corrected.
Introduction
C: Line 38: needs grammatical correction
A: All grammatical errors have been corrected (Line 44)
C: Line 46: Please change assume to past tense (assumed)
A: Done (Line 52)
C: Lines 55-56: Note: periods missing at end of sentence
A: We have revised punctuation errors
C: Line 57-61: sentences need grammatical correction
A: This paragraph has been rephrased (Line 76-78)
C: Line 66: please correct “these” to “this” alternative bracket placement procedure. Similarly, line 67
A: Done
Materials and Methods
C: Lines 81-82 need grammatical correction
A: Done (Lines 101-103)
C: Line 82: Please note typo: “Fort he” Please define abbreviations in the Tables
A: Amended (Line 10)
C: Line 147: please correct grammatically
A: Done
C: Furthermore, please note the following major methodological concerns:
There is no mention of how allocation of teeth to the study groups were performed (for instance ideally there should have been randomization).
Ideally, assessment of the outcome should have been blinded to minimize biases. The lack of randomization and blinding should be addressed; and limitation should be presented in the Discussion section.
A: Thank you for addressing this methodological issues. We have added the clarification that the samples were randomly allocated in each group (Line 98)
The lack of blinding is addressed in the discussion. We tried to solve this limitation by the thorough examination of an experience orthodontist of the samples, with the supervision of a senior researcher in the field (Lines 288-291)
Statistical analysis
C: Since multiple group comparisons were performed; post hoc testing (such as Bonferroni test) is needed after ANOVA to assess differences between each group pair and to control errors.
A: Bonferroni test was performed on both cases (SBS and ARI comparisons). However we didn’t address this results in the initial manuscripts. Thanks to your observations, we have added this data to make the statistical analysis information more extensive.
C: Also, what is the reliability in the measurements? Were the intra- and inter-observer reliabilities examined?
A: Unfortunately, once debonded, intra operator tests can not be repeated on the shear bond strength values. On the other hand, intra observer test were indeed performed for the adhesive remnant index samples; and we have added this information in the statistical analysis´ sub-heading
C: Furthermore, how was the sample size estimated? Ideally, a priori power analysis is needed to calculate the required sample size. Lack of a power analysis is a major limitation.
A: We conducted a previous pilot study to test the methodology and then performed a sample size calculation with a power set at 80% before the research. Thank you very much for addressing this matter. We have added a sample size subheading in the material and methods section to clarify this point.
Results
C: Pairwise comparisons need to be performed and p-values need to be adjusted when ANOVA is significant. For instance, it is mentioned that: “The results showed statistically significant differences between the three types of adhesive systems.” Further testing is needed for group- pair comparisons; and adjusted p-values need to be provided for each group-pair comparison.
A: Tables 5 and 6 have been modified adding p-value data of all group-pair comparisons
C: Similarly, detailed p-values need to be reported for all outcomes including the adhesive remnant index.
A: We have also add this data to table 7
Discussion
C: Line 220: Sentence needs grammatical correction.
A: Done (Line 288-292)
Lines 243-4: Please note that grammatical correction is needed.
A: These lines have been rephrased (Line 322-324)
C: Line 283: needs grammatical correction
A: Done 353-356
C: Need to further address potential limitations; such as lack of power, potential biases and error in measurements. Also, since only premolars were examined this could affect generalizability of the results. The generalizability and clinical implications of this study need to be addressed.
A: We have extended the limitations section in the discussion addressing all of this subjects; and also explained our rationale for premolar selection (Lines 292-296).
Conclusions:
C: Needs grammatical correction, also please use abbreviation as used previously in the text.
A: All conclusions have been rephrased to provide a clearer statement.
PD: Please find attached the revised version of the manuscript

Reviewer 3 Report
1) Justify in the annotation the choice of bonding systems Transbond XT ® , Beauty Ortho Bond ® and GC Ortho Connect ® and decipher their abbreviations. 2) in the abstract, in the results section, specify all comparative data, and in the conclusions section, specify numbers instead of the words "significantly affects". 3) the abstract should show the further direction of research. 4) the manuscript should emphasize the choice of systems Transbond XT ® , Beauty Ortho Bond ® and GC Ortho Connect ® . 5) for the purpose of clarity of the material for readers, photos of experimental samples and processes for measuring the strength and glue residues are required. 6) Remove typos (points) in the subheadings "Debonding resistance testing" and "Statistical analysis". 7) rewrite the Conclusions taking into account the novelty, quantitative comparative data and prospects for further research.Author Response
Dear Reviewer, we appreciate all your observations and want to thank you for your thorough examination of our manuscript. We have made an extensive revision of the text, with special attention to the matters you kindly pointed out. We hope our new version meets the journals high quality standards.
C:Justify in the annotation the choice of bonding systems Transbond XT ® , Beauty Ortho Bond ® and GC Ortho Connect ® and decipher their abbreviations.
A: Justification for the choice of systems has been pointed out both in the introduction and discussion and abbreviation errors have been revised
C: In the abstract, in the results section, specify all comparative data, and in the conclusions section, specify numbers instead of the words "significantly affects".
A: We have added more data to the results in the abstract and have made improvements on the conclusions.
C: The abstract should show the further direction of research.
A: Further in vivo studies are indeed needed and we point out this matter in the abstract
C: The manuscript should emphasize the choice of systems Transbond XT ® , Beauty Ortho Bond ® and GC Ortho Connect ®
A: Adhesive systems choice is addressed both in introduction and discussion sections (Lines 75-78; 306-313)
C: For the purpose of clarity of the material for readers, photos of experimental samples and processes for measuring the strength and glue residues are required.
A: Figures 1 and 2 have been added to clarify sample processing information.
C: Remove typos (points) in the subheadings "Debonding resistance testing" and "Statistical analysis".
A: Done
C: Rewrite the Conclusions taking into account the novelty, quantitative comparative data and prospects for further research.
A: We have re-wrote the conclusions to highlight most interesting data of our research.
PD: Please find attached the revised version of the manuscript

Round 2
Reviewer 1 Report
Dear authors,
I am very sorry to ascertain that no significant changes were made to the article and no real improvement can be observed.
You did not care to follow the template and the instructions for authors when revising the manuscript.
Basic issues, as giving the manufacturers name and location were not attended.
For future knowledge, scientific or academic titles do not belong to the affiliation and should not be mentioned.
Language mistakes are still present.
Author Response
Dear reviewer, we regret you still consider our work has not been improved. As stated by the other two reviewers, we have performed an extensive revision and almost all of the comments and suggestions have been addressed properly. We are once again presenting our paper for your consideration, clarifying a few additional issues of concern.
We receive your comments about our work with disappointment. We have put a great amount of effort in improving our manuscript according to your first review of which we are most grateful for. However, we continue to state that as far as we know, there is no study alike ours that compares these two novel materials. Novel materials testing is a very interesting subject, specially from the clinical point of view and therefore we wish to continue clinical trials in this line of research. Many practitioners would benefit of a bonding protocol with the reduction of clinical steps, and our study has shown that the most reliable bonding procedures are still acid-etching dependent.
Concerning your specific remarks:
The manufactures name and location have been added.
Affiliations have been corrected.
Spelling errors have been revised again.
For this reasons, we kindly ask you to review our work again.

Reviewer 2 Report
Dear Authors,
Thank you for the extensive revisions. Most of my concerns/suggestions have been addressed in the revised version; and I believe the quality of the manuscript has been considerably improved.
However, please note that further editing is needed for some grammatical corrections and English editing throughout the manuscript. Furthermore, I have a serious concern with the results [Line 230: …“whereas the bonding technique does not significantly influences bonding strength (p=0.04).”]
This statement is strange since it mentions no significant influence; yet the p value is less than 0.05. Please clarify as this affects the main results and conclusions of you research.
Author Response
Dear reviewer, we are very grateful for all of your comments and extensive revision of our manuscript.
Thanks to your concerns about our statistical data and conclusions, we decided to re-do all statistics. For this matter, we decided to perform a one-way ANOVA to better compare values of SBS in the six groups of study.
In the previous ANOVA test showed in table 5, bonding technique results were very close to a value of “no statistical significance >0.05 and we feared an inadequate correction had been performed for this ANOVA.
The new analysis of bonding technique with t-test are much accurate on this matter and revealed a p = 0.160 when comparisons of direct and indirect bonding were performed. This data is shown in new table 5.
These results coincide with our original conclusions.
